# Tau protein aggregation associated with SARS-CoV-2 main protease

Raphael Josef Eberle[1,2]*, Mônika Aparecida Coronado[1], Ian Gering[1], Simon Sommerhage[1], Karolina Korostov[1], Anja Stefanski[3], Kai Stühler[3], Victoria Kraemer-Schulien[1], Lara Blömeke[1,2], Oliver Bannach[1,2,4], Dieter Willbold[1,2,5]

**1** Institute of Biological Information Processing (IBI-7: Structural Biochemistry), Forschungszentrum Jülich, Jülich, Germany, **2** Institute of Physical Biology, Heinrich-Heine-University Düsseldorf, Düsseldorf, Germany, **3** Molecular Proteomics Laboratory (MPL), BMFZ, Heinrich-Heine-University Düsseldorf, Düsseldorf, Germany, **4** attyloid GmbH, Düsseldorf, Germany, **5** JuStruct: Jülich Centre for Structural Biology, Forschungszentrum Jülich, Jülich, Germany

* r.eberle@fz-juelich.de

**Data Availability Statement:** All relevant data are within the paper and its Supporting information files.

## Abstract

The primary function of virus proteases is the proteolytic processing of the viral polyprotein. These enzymes can also cleave host cell proteins, which is important for viral pathogenicity, modulation of cellular processes, viral replication, the defeat of antiviral responses and modulation of the immune response. It is known that COVID-19 can influence multiple tissues or organs and that infection can damage the functionality of the brain in multiple ways. After COVID-19 infections, amyloid-β, neurogranin, tau and phosphorylated tau were detected extracellularly, implicating possible neurodegenerative processes. The present study describes the possible induction of tau aggregation by the SARS-CoV-2 3CL protease (3CL^pro) possibly relevant in neuropathology. Further investigations demonstrated that tau was proteolytically cleaved by the viral protease 3CL and, consequently, generated aggregates. However, more evidence is needed to confirm that COVID-19 is able to trigger neurodegenerative diseases.

## Introduction

Viral pathogens encode their protease(s) or use host proteases for their replication cycle. In the case of acute respiratory syndrome coronavirus 2 (SARS-CoV-2), proteolytic cleavage of the two virus polyproteins generates the various viral proteins needed to form a replication complex required for transcription and replication of the viral genome and subgenomic mRNAs. The key viral enzymes responsible are the papain-like (PLP, nsp3) and 3-chymotrypsin-like proteases (3CL^pro) [1–3]. In addition, host cell protein cleavage is a critical component of viral pathogenicity [4], including diverting cellular processes to viral replication, defeating antiviral responses and immune response modulation. Many large-scale analyses of the SARS-CoV-2 infected-cell transcriptome, proteome, phosphoproteome and interactomes are described [5–7]. Regarding the 3CL^pro human substrate repertoire, also known as the degradome [8], Pablos et al., 2021 identified over 100 substrates and 58 additional high confidence candidate substrates out of SARS-CoV-2 infected human lung and kidney cells [9].

**Funding:** D.W. is supported by Deutsche Forschungsgemeinschaft (DFG, German Research Foundation), Project-ID 267205415, SFB 1208. sFIDA was supported by the programs "Biomarkers Across Neurodegenerative Diseases I + II" of The Alzheimer's Association, Alzheimer's Research UK and the Weston Brain Institute (11084 and BAND-19-614337). We are also grateful for support from The Michael J. Fox Foundation for Parkinson's Research (14977, 009889), from the ALS Association and from the Packard Center (19-SI-476). We received further funding from the Deutsche Forschungsgemeinschaft (INST 208/616-1 FUGG, INST 208/794-1 FUGG) and the Helmholtz Association (HVF0079).

**Competing interests:** The authors have declared that no competing interests exist.

SARS-CoV-2 was identified in December 2019 as the causative agent of coronavirus disease-19 (COVID-19) first occurring in Wuhan, Hubei province, China [10]. According to the data that were reported to the World Health Organization (WHO) up to the 3$^{rd}$ of January 2023, the global SARS-CoV-2 pandemic was associated with >655 million confirmed cases of infections and >6.6 million virus-related deaths worldwide [11].

It is well known that SARS-CoV-2 infection can influence multiple tissues or organs [12–16]. Post-COVID syndrome (also known as Long-COVID) has also been described as a syndrome that encompasses a prolonged course of various physical and neuropsychiatric symptoms that persist for more than 12 weeks [17, 18]. It has also been reported that COVID-19 can damage the brain in different ways (Table 1).

Douaud et al. 2022 described the dramatic effects of SARS-CoV-2 infections on the brain structure, including a reduction in grey matter thickness, tissue damage in regions that are functionally connected to the primary olfactory cortex and a significant reduction in global brain size [28]. However, so far, there exists no direct link with the generation of neurodegenerative diseases like Parkinson's disease (PD), Alzheimer's disease (AD), amyotrophic lateral sclerosis (ALS), frontotemporal dementia (FTD), Huntington's disease (HD), spinocerebellar ataxias, corticobasal degeneration, progressive supranuclear palsy, chronic traumatic encephalopathy, or multiple system atrophy. These diseases have many common features, including their chronic and progressive nature, the increased prevalence with age, destruction of neurons in specific areas of the brain, damage to the network of synaptic connections, and selective brain mass loss [29]. Another event is the progressive accumulation of misfolded protein aggregates with well-ordered structures. The proteins most commonly implicated in the accumulation of cerebral misfolded aggregates include amyloid-beta (Aβ), tau, alpha-synuclein (α-Syn) and TAR DNA-binding protein 43 (TDP-43) [29].

After COVID-19 infections in the brain, amyloid-β, neurogranin, tau and phosphorylated tau can be detected extracellularly, implicating possible neurodegenerative processes [30]. Another study demonstrated that the spike protein receptor binding domain binds to heparin and heparin-binding proteins, including amyloid-β, α-synuclein, tau, prion and TDP-43, which may initiate the pathological aggregation of these proteins resulting in neurodegeneration [31, 32]. Ramani et al. 2020 showed that SARS-CoV-2 targets neurons of 3D human brain organoids and neurons invaded with SARS-CoV-2 at the cortical area display altered tau, tau hyperphosphorylation distribution and apparent neuronal death [33].

Tau phosphorylation and tau proteolysis are likely key factors in disease-associated tau aggregation and accumulation. Tau proteolysis can destabilise its primary structure, preventing correct folding and can lead to the formation of aggregated tau species due to a disordered quaternary structure. Tau can be cleaved by various proteolytic enzymes, including caspases,

**Table 1. Some neurological symptoms caused by COVID-19.**

| Neurological symptom | Reference |
| --- | --- |
| Loss of smell (anosmia) and altered taste (ageusia) | [19] |
| Myoclonus, cerebellar ataxia, seizure and tremor | [20, 21] |
| Headache | [22] |
| Cardiorespiratory failure | [23] |
| Encephalopathy | [24] |
| Acute Disseminated Encephalomyelitis | [25] |
| Stroke | [26] |
| Guillain-Barre syndrome | [27] |

calpains, thrombin, cathepsins, metalloprotease 10, asparagine endopeptidase and puromycin-sensitive aminopeptidase [34].

Here, we report the cleavage and aggregation of tau after SARS-CoV-2 3CL$^{pro}$ treatment *in vitro* using a combination of ThT assays, negative staining TEM, analytical HPLC and mass spectrometry.

## Material and methods

### Preparation of alpha-synuclein, TDP-43 and 2N4R tau

Alpha-synuclein was cloned, expressed and purified, as described previously [35]. TDP-43 sample was kindly provided by Dr Jeanine Kutzsche (IBI-7, Forschungszentrum Jülich). The gene for human tau (2N4R isoform, uniprot ID: P10636-8) encodes a protein of 441 amino acids. The respective gene was commercially synthesised and cloned into the pET28A(+) vector (Genentech, San Francisco, USA), without His-tag. Protein expression was performed as described previously [36].

Protein extraction began by dissolving the cell pellet of 1 L expression in 30 ml buffer 1 (50 mM HEPES pH 7.5, 500 mM KCL, 5 mM β-ME and 1 mM EDTA). The dissolved cell pellets were heated for 30 min at 85˚C followed by 10 min on ice, and samples were sonicated 3 x 40 seconds at a power setting of 5 in an ultrasonic cell disruptor Modell 250 (Branson Ultrasonic, Brookfield, USA). Bacterial debris was pelleted for 50 min at 10,000 x g. Soluble tau protein was precipitated from the supernatant by adding 40 ml of a saturated ammonium sulfate solution and incubated for 30 minutes at room temperature. Afterwards, the samples were centrifuged for 30 min at 10,000 x g, and the pellet was resuspended in buffer 2 (50 mM HEPES pH 7.5, 50 mM KCL, 1.5 M ammonium sulfate, 2 mM TCEP and 1 mM EDTA). The solution was centrifuged for 30 minutes at 10,000 x g, and the pellet was resolved in 10 mL ddH$_2$O and 2 mM TCEP. The sample was centrifuged again for 30 minutes at 10,000 x g, and the appearing pellet was resuspended in buffer 3 (20 mM HEPES pH 6.7, 150 mM NaCl, 2 mM TCEP and 1 mM EDTA). The sample was centrifuged for 1h at 12,000 x g, and the supernatant was dialysed overnight at 6˚C against buffer 4 (50 mM ammonium acetate pH 7.4, 1 mM TCEP). The protein purity was assessed by SDS/PAGE (15%).

### Western blot analyses

For western blots, a fluorescent anti-tau antibody was used. Therefore tau13 (Biolegend, San Diego, USA) was labelled with CF633 (Biotium, Freemont, USA), and the labelling process was performed as described previously [37]. 2N4R tau recombinant protein samples were prepared in Laemmli buffer (final 1× composition: 20 mM Tris, pH 6.8, 2% SDS, 6% glycerol, 1% β-ME, 0.002% Bromophenol Blue). All samples were heated at 95˚C for 5 min and separated using SDS PAGE (15%). Proteins were then transferred to a polyvinylidene fluoride (PVDF) membrane (Thermo Fisher Scientific, Waltham, USA) at 500 mA for 40 min. After a washing step for 15 min in Tris-buffered saline tween buffer (TBS-T) (20 mM Tris, 150 mM NaCl, 0.1% Tween 20), the membrane was blocked for 1 h with 2.5% milk powder/TBS-T. Next, the membrane was washed with TBS-T, 2 × 5 min and in the last step for 15 min. Tau13 stocks were 1 mg/ml and were diluted in TBS-T (1:5000). The membrane was incubated with the antibody for 1.5 h. at RT. After a final wash step (2 × 5 min and 1 × 10 min), TBS-T was performed. Detection based on the CF633 fluorescence of the labelled tau13 antibody. Bio-Rad universal hood II and Chemidoc XRS camera and Quantity One 4.6.5 software enabled the visualisation and quantification of the protein bands.

## Cloning, expression and purification of SARS-CoV-2 3CL^pro

SARS-CoV-2 3CL$^{pro}$ (Uniprot entry: P0DTD1, virus strain: hCoV-19/Wuhan/WIV04/2019) was cloned, expressed and purified, as described previously [38].

## Thioflavin T aggregation (ThT) assay

ThT aggregation assays were conducted in Corning half area 96-well plates with the non-binding surface (Corning No. 3881, Glendale, AZ, USA). As a control, polymerisation of 2N4R tau was initiated in the presence of the aggregation inducer heparin (Sigma-Aldrich, USA) with a molar ratio of 4:1 (Tau:heparin). 10 μM tau was incubated with 2.5 μM heparin (the final volume of the reaction mixture was 150 μl). The experiment buffer contained 20 mM Tris pH 7.2, 200 mM NaCl, 1 mM TCEP and 10 μM ThT. Fluorescence intensities were measured at 6 minutes intervals over 30 hours at 350 rpm and 37°C using an Infinite 200 PRO plate reader (Tecan, Männedorf, Switzerland). The excitation and emission wavelengths were 440 and 490 nm, respectively. All measurements were performed in triplicate, and data are presented as mean ± SD.

## ThT assay using SARS-CoV-2 3CL^pro as tau aggregation inducer

ThT assays were performed as described before. Instead of heparin, 10 μM SARS-CoV-2 3CL$^{pro}$ was used as an aggregation inducer. In a preliminary test, the effect of 3CL$^{pro}$ against 10 μM 2N4R tau, α synuclein and TDP-43 was tested over 24h. As a control, the same experiment was performed with the single proteins.

A further experiment was performed with inactivated 3CL$^{pro}$, the protease was incubated with an equimolar concentration of Disulfiram (DSF) (10μM) for 30 minutes at RT, after 10 μM 2N4R tau and ThT was added, and the experiment was running for 24h. As a control, 2N4R tau was incubated with DSF and monitored for the same experimental time.

Furthermore, a ThT assay of 10 μM tau and 3CL$^{pro}$ were stopped after 24h, and the protease was inactivated through DSF addition (10 μM) and incubated for 30 minutes at RT. Afterwards, a new tau sample (10μM) was added. The same procedure without inactivating the protease was followed as a control. All measurements were performed in triplicate, and data are presented as mean ± SD.

## Circular dichroism (CD) spectroscopy

CD measurements were carried out with a Jasco J-1100 Spectropolarimeter (Jasco, Germany). Far-UV spectra were measured in 190 to 250 nm using a 2N4R tau (monomer and filament) concentration of 10 μM in 20 mM $K_2HPO_4/KH_2PO_4$ pH 7.0.

2N4R tau aggregates were generated in the presence of heparin (Sigma-Aldrich, USA) with a molar ratio of 4:1 (Tau:heparin). 10 μM tau was incubated with 2.5 μM heparin. 2.5 μM heparin was measured as blank sample and the signal was subtracted from the tau signal. As an additional control the CD spectrum of 25 μM heparin was collected.

A 1 mm path length cell was used for the measurements; 10 repeat scans were obtained for each sample, and five scans were conducted to establish the respective baselines. The averaged baseline spectrum was subtracted from the averaged sample spectrum. The results are presented as molar ellipticity [θ], according to the Eq (1):

$$[\theta]\lambda = \theta/(c*0.001*l*n) \tag{1}$$

where θ is the ellipticity measured at the wavelength λ (deg), c is the protein concentration (mol/L), 0.001 is the cell path length (cm), and n is the number of amino acids.

## Investigation of tau and 3CL^pro doses dependency on the aggregation process

Different concentrations were titrated to investigate tau, and 3CL^pro doses dependency on the 2N4R tau fibril formation and a ThT assay was performed as described before. The effect of 0, 2.5, 5, 10, 25 and 50 μM 3CL^pro was tested against 10 μM tau over 24h. We also tested the opposite effect, where 0, 5, 10, 20, 40, 60, 80, and 100 μM of tau were tested against 10 μM 3CL^pro. All experiments were performed in triplicate, and data are presented as mean ± SD.

## Stability of tau in the presence of Sars-CoV-2 3CL^pro investigated by High-performance liquid chromatography (HPLC)

To explore the proteolytic degradation of tau by 3CL^pro samples after 0, 24, 48 and 72 h incubation were analysed by high-performance liquid chromatography (HPLC). After incubation the samples were centrifuged for 60 minutes at 10,000 x g, to separate aggregated and soluble tau fragments.

Agilent 1260 Infinity II system (Agilent Technologies, Santa Clara, CA, USA), equipped with a quaternary pump, autosampler, heated column compartment, multi-wavelength detector (MWD) and an analytical fraction collector, was used. 20 μL of sample solution was injected into an Agilent Zorbax 300-SB C8 4.6*250 mm, 5 μm reversed-phase liquid chromatography column (Agilent Technologies, Santa Clara, CA, USA), which was heated to 80˚C. Mobile phases consisted of A: Water + 0.1% Trifluoroacetic acid (TFA) and B: Acetonitrile + 0.1% TFA. Analyte elution was accomplished by a linear gradient from 10% to 80% buffer B in 20 min. Chromatograms were acquired at 214 nm and 280 nm. Furthermore, data acquisition and evaluation were performed with the Agilent OpenLab software (Version 2.6). The mean peak area of each triplicate was plotted against incubation time. Chromatograms of tau with 3CL^pro and chromatograms with a single protein were investigated for metabolite formation after an incubation period of up to 72 h. The peaks related to the sample in which both proteins were present (max incubation 72 h) were considered potential metabolites of the tau protein produced by Sars-CoV-2 3CL^pro.

## Purification of tau metabolites after proteolytic degradation by Sars-CoV-2 3CL^pro

To further investigate tau metabolites produced by incubation with Sars-CoV-2 3CL^pro, we used HPLC and further mass spectrometry (MS) analysis. Tau was incubated with the protease for 72h at 37˚C and 500 rpm. HPLC conditions were the same as described above. 100 μL of the sample was applied to the column per each chromatography run, and fractions were collected every minute. Fractions containing the same peak were pooled and lyophilised; subsequently, the samples were submitted for MS analysis.

## Sample processing and mass spectrometry

Lyophilised samples were resuspended in 500 μL 0.1% trifluoroacetic acid (TFA) and digested with trypsin (Serva, Heidelberg, Germany) in 50 mM $NH_4HCO_3$ overnight at 37˚C. Tryptic peptides were extracted with 0.1% TFA and subjected to MS-coupled liquid chromatography. Briefly, for peptide separation over a 55-minute LC-gradient with 300 nL/min in an Ultimate 3000 Rapid Separation liquid chromatography system (Thermo Scientific, Bremen, Germany) equipped with an Acclaim PepMap 100 C18 column (75 μm inner diameter, 25 cm length, 2 mm particle size from Thermo Scientific, Bremen, Germany) was used. MS analysis was carried out on a Q-Exactive plus mass spectrometer (Thermo Scientific, Bremen, Germany)

operating in positive mode and equipped with a nanoelectrospray ionisation source. The capillary temperature was set to 250˚C and the source voltage to 1.5 kV. Survey scans were carried out over a mass range from 200–2,000 m/z at a resolution of 70,000 (at 200 m/z). The target value for the automatic gain control was 3,000,000, and the maximum fills time was 50 ms. The 20 most intense peptide ions (excluding singly charged ions) were selected for fragmentation. Peptide fragments were analysed using a maximal fill time of 50 ms, automatic gain control target value of 100,000 and a resolution of 17,500 (at 200 m/z). Already fragmented ions were excluded for fragmentation for 10 seconds.

Acquired spectra were searched using Sequest HT within Proteome Discoverer version 2.4.1.15 against the SwissProt *Homo sapiens* proteome dataset (UP000005640, 75777 sequences) with the inserted sequence of the human tau protein and an *E.coli* BL21 (DE3) database (UP000002032, 4156 sequences). Methionine oxidation was considered a variable modification and tryptic cleavage specificity with a maximum of two missed cleavage sites. For the main search, a precursor mass tolerance of 10 ppm and a mass tolerance of 0.02 Da were applied for fragment spectra. For the semi-specific tryptic search of peptides, PEAKS Studio 10.6 Build 220201221 was used, and the above human database was searched with an error tolerance of 20 ppm for parent masses and an error tolerance of 0.2 Da for fragment masses.

## Surface-based fluorescence intensity distribution analysis

To quantify tau aggregation due to 3CL^pro activity, surface-based fluorescence intensity distribution analysis (sFIDA) was performed according to the biochemical principle of Kravchenko et al. 2017, and Herrmann et al. 2017 [39, 40]. Therefore, we used 384-Well plates (Greiner, Kremsmünster, Austria) to incubate the capture-antibody tau12 in 0.1 M carbonate at a 2.5 μg/ml concentration. After a fivefold washing step with TBS-T and TBS, 80 μl of blocking solution (Candor Bioscience, Wangen, Germany) was added and incubated for 1.5 h. Afterwards, the plate was washed like previously described, and 20 μl of tau conjugated silica-nanoparticles (SiNaPs) and tau aggregates diluted in low cross buffer strong (Candor Bioscience GmbH, Wangen in Allgäu, Germany) were added. In Addition, 20 μl of 10 nM and 100 nM tau monomer were applied.

To investigate the formation of tau oligomers or aggregates induced by 3CL^pro, 5 μM tau monomer was incubated with 5 μM 3CL^pro for 72 h. The negative control was established equivalently; however, the protease was previously inactivated with 10 μM Disulfiram (DSF).

The samples, buffer control (BC) and capture control (CC), were incubated for 2 h and washed five times with TBS. For capture control, the capture antibody was omitted. As a detection probe, 20 μl of 0.078 μg/ml tau13 CF633 in TBS was used. After 1 h of incubation, the wells were washed five times with TBS, and the buffer was changed against TBS-ProClin. The SiNaPs and antibodies used were synthesised and labelled according to the previously described principle [37, 41].

## Negative staining transmission electron microscopy (TEM)

Tau samples were prepared as follows, 20 μM tau and 10 μM 3CL^pro were incubated for 72 h at 37˚C. To prepare specimens for negative staining TEM, a total volume of 3 μL from the corresponding sample was pipetted onto carbon coated 300 Mesh copper EM grids, which were operated by anti-capillary tweezers and glow discharged in advance for 90 seconds using a PELCO easiGlow™ (Ted Pella, Inc., Redding, California, USA) at 0.39 mbar and 15 mA. After an incubation time of 1 min excess liquid was blotted away by touching the edge of the grid with filter paper. Subsequently the grids were stained with two 4 μL droplets of a 2% Uranyl acetate solution. While the first droplet's excess liquid was immediately blotted after

application, the second droplet stayed on the grid for an incubation time of 1 min before blotting. Finally, the grids were left to dry at room temperature for approximately 1 min. These negatively stained samples were examined on a Talos L120C G2 transmission electron microscope (Thermo Fisher Scientific, Waltham, Massachusetts, USA) which was operated at 120 kV (LaB6 (Lanthanum hexaboride)/Denka). Micrographs were collected in medium magnification (6700x) and high magnification (73kx) on a 4k x 4k Ceta 16M CEMOS camera using the Velox™ software (Thermo Fisher Scientific, Waltham, Massachusetts, USA).

## Results and discussion

### Purification of 2N4R tau with a precipitation approach and characterisation of the protein

2N4R tau was expressed in BL21 (DE3) (T1) *E. coli* and purified by a precipitation approach. 2N4R tau consists of 441 amino acids with an approximated molecular weight of 46 kDa. The purity was assessed by SDS PAGE (S1 Fig). However, the protein presented a single band on a denaturing SDS-PAGE gel with an apparent molecular mass of around 67 kDa, this behavior was described previously [42, 43]. A western blot with the specific antibody (Tau13, Biolegend) confirmed the target protein (S2A Fig). Following successful purification, 2N4R tau was characterised to compare the properties to those previously reported [44–47]. It is well known that tau, in the monomeric state, is inherently unfolded, with predominantly random-coil conformation. Our CD analysis confirmed this observation for the purified protein, with minimum peaks around 200 nm (S2B Fig). Tau aggregation was investigated using ThT assay and heparin as an inducer [48]. The results of the ThT assay indicated that heparin promoted the induction and acceleration of tau aggregation within 24h (S2C Fig). The structural changes of tau in the presence of heparin were followed by CD spectroscopy, demonstrating a shift of the absorbance spectrum from 202 (random-coil conformation) to 213 (beta-sheet conformation) nm (S2B Fig). As a control, the CD spectrum of 2.5 μM heparin was measured, the spectra showed no specific minimum at 213 nm (S3B Fig). At a heparin concentration of 25 μM, the CD spectrum showed a minimum at around 220 nm (S3B Fig). However this heparin concentration was not reached during the experiment. Those results demonstrated that tau had been successfully aggregated by heparin because of the characteristic transition from a random coil to beta-sheet conformation in the tau secondary structure [45].

### Identification and characterisation of tau aggregation events induced by SARS-CoV-2 3CL$^{pro}$ using ThT assay

To identify a possible aggregation effect caused by SARS-CoV-2 3CL$^{pro}$ on tau, alpha-synuclein and TDP-43 proteins, a ThT fibrillation assay was performed. The purity of alpha-synuclein and TDP-43 was assessed by SDS PAGE (S4 Fig). The preliminary test showed that tau aggregates increased over time after the addition of 10 μM of SARS-CoV-2 3CL$^{pro}$, which was not observed for alpha-synuclein and TDP-43 (Fig 1A). To confirm our preliminary results, control experiments using the single proteins was performed, which indicated no signal of aggregation over the time (Fig 1B).

Additional ThT experiments were performed to evaluate the dose dependency of SARS--CoV-2 3CL$^{pro}$ and 2N4R tau concentration on the aggregation behaviour of tau. A higher concentration of the protease (0, 0.5, 1, 2.5, 5 and 10 μM) caused a higher amount of the tau aggregates in a given time of 24h (S5A Fig). Similarly, the titration of tau at different concentrations (0, 5, 10, 20, 40, 60, 80 and 100 μM) resulted in a dose-dependent manner to a higher amount of aggregates (S5B Fig).

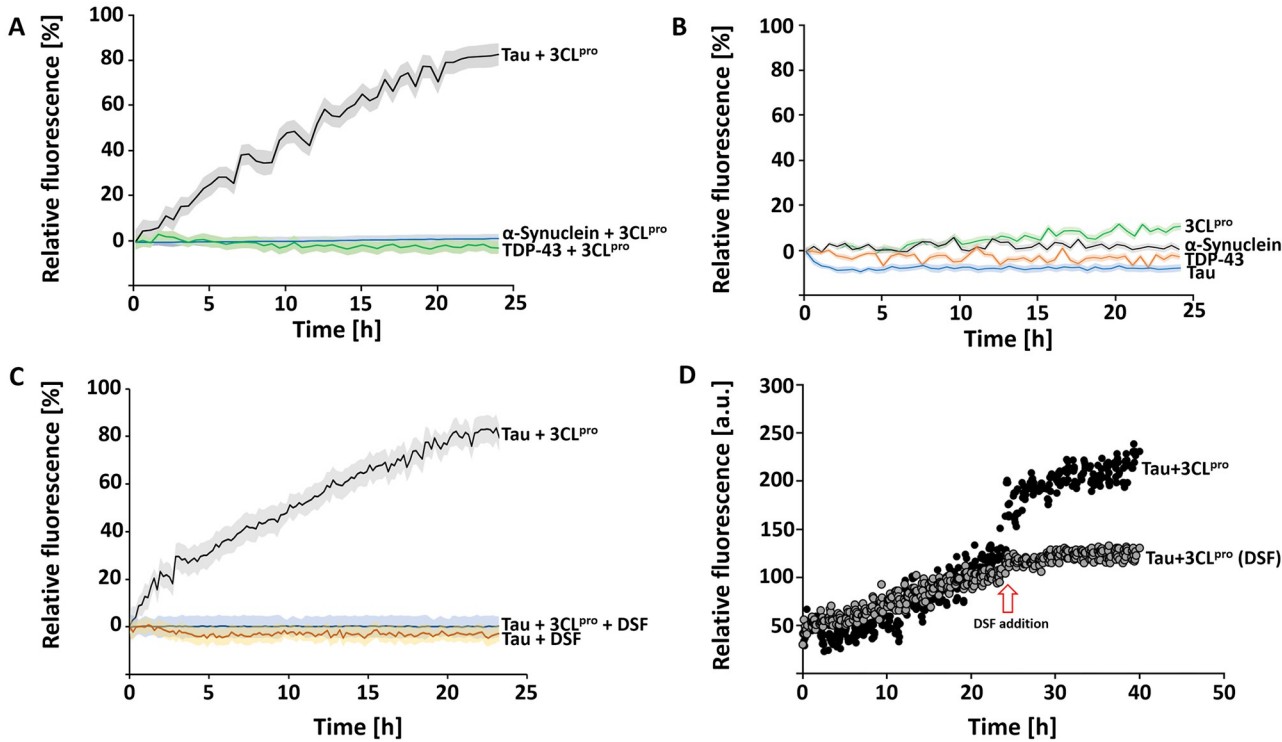

**Fig 1. Effect of SARS-CoV-2 3CL^pro on 2N4R tau aggregation and effect of 3CL^pro inactivation on tau aggregation. A**: ThT assay of 2N4R tau, α-Synuclein and TDP-43, 3CL^pro used as aggregation inducer. The experiment was performed for 24 h, 37°C and 600 rpm. **B**: Control ThT assay of single 3CL^pro, α-Synuclein, TDP-43 and tau. **C**: ThT assay of inactivated 3CL^pro, tau aggregation was not observed. **D**: ThT assay of 2N4R tau, aggregation was induced by 3CL^pro. After 24h DSF inactivated the protease and after 30 minutes of incubation, 10 μM fresh tau was added. Data shown are the mean ± SD from three independent measurements (n = 3).

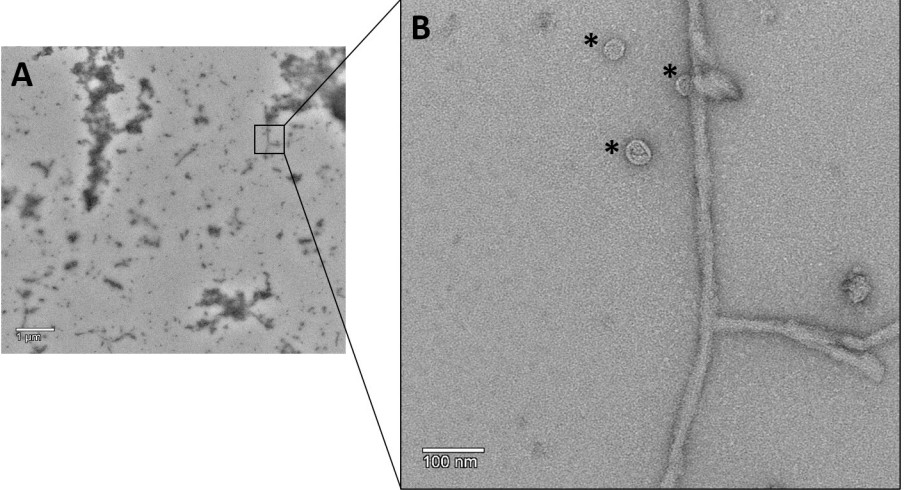

**Fig 2. Negative stain TEM of 3CL^pro induced tau fibrils. A**: Representative micrograph at medium (6700x) magnification. **B**: Representative micrographs at high (73kx) magnification. Asterisks label vesicle structures.

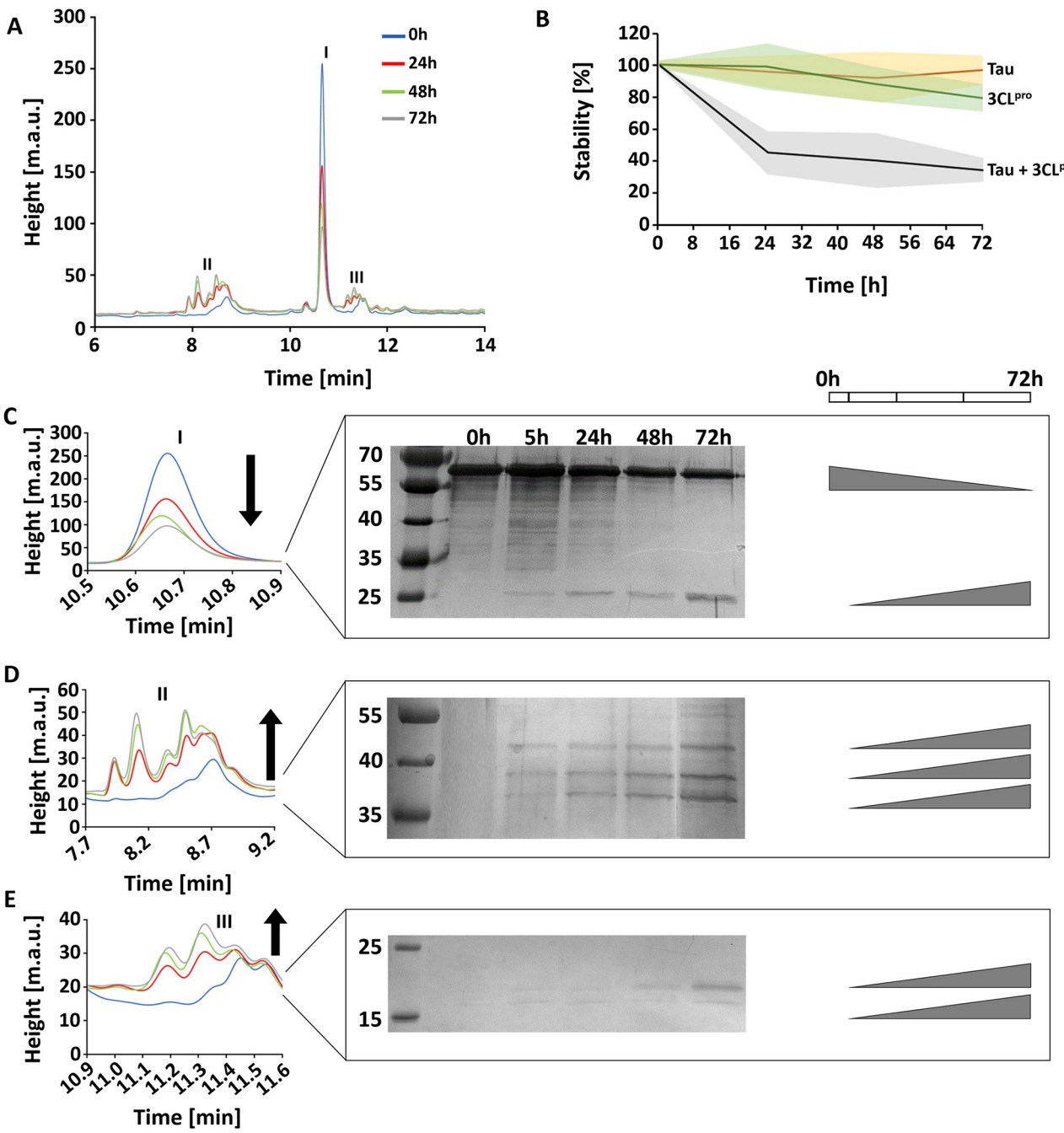

**Fig 3. Effect of SARS-CoV-2 3CL^pro on 2N4R tau degradation. A**: Analytical HPLC analysis of 2N4R tau incubated with SARS-CoV-2 3CL^pro for 0, 24, 48 and 72h. The corresponding chromatogram regions of the tau monomer and related metabolites are labelled (I-III). **B**: Stability of 2N4R tau monomer after treatment with 3CL^pro over 72h. Single tau and 3CL^pro are shown as control. After 3CL^pro treatment, the tau monomer amount decreases by about 60%. **C**: Chromatogram of peak I (Tau) shown enlarged, a silver stained SDS PAGE demonstrated that the 2N4R tau amount decreased over 72h treatment with 3CL^pro and a protein band increases at 25 kDa. **D**: Chromatogram of peak region II shows enlarged three protein bands appearing over 72h experimental time. **E**: Chromatogram of peak region III shown enlarged, two protein bands appear over the 72h experimental time.

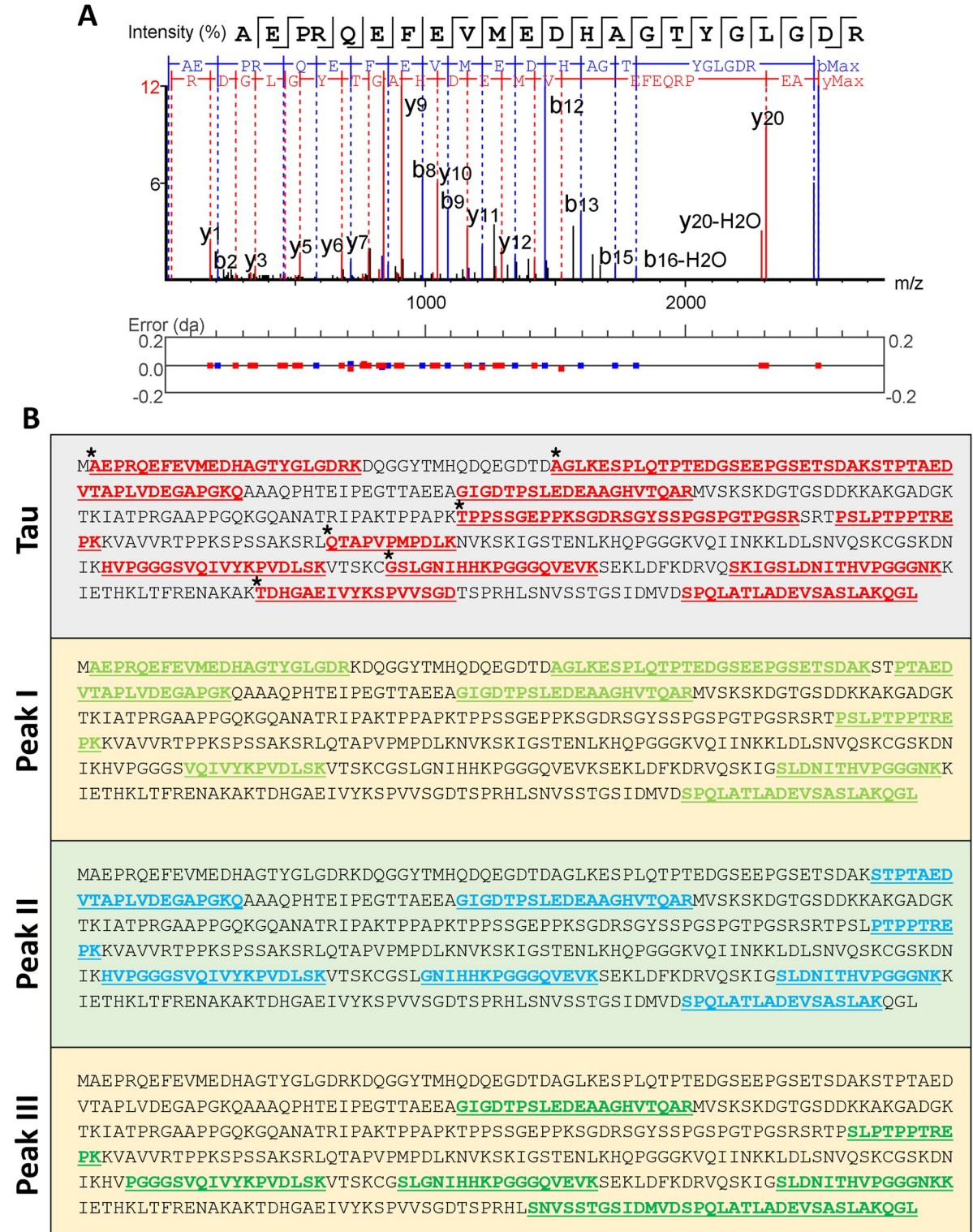

**Fig 4. Mass spectrometry analysis of tau metabolites. A**: Example of an MS spectrum corresponding to a tryptic peptide derived from 2N4R tau (AEPRQEFEVMEDHAGTYGGLGDR), b- and y-ions are labelled. **B**: 2N4R tau sequences and tryptic peptides, which could be identified in the corresponding samples, are highlighted. Asterisks label tryptic 2N4R tau peptides, which could not be identified in the three peak samples.

To extend our analysis of the tau aggregation process induced by 3CL$^{pro}$, negative staining transmission electron microscopy (TEM) was performed on solutions containing tau aggregates after treatment with the protease. The analysis demonstrated that the solutions contain amorphous aggregates and fibrils (Fig 2A and 2B and S6 Fig). The tau fibrils morphology is similar to those described in previous studies [44–47].

Besides the tau fibrils, micelle or vesicle-like structures were observed (Fig 2B). As described before, anionic micelles or vesicles can serve to nucleate tau fibrillation [49, 50], which may explain the observation of the vesicle-like structures in the negative stain TEM micrographs.

## Effect of SARS-CoV-2 3CL$^{pro}$ inactivation on tau aggregation

There exist two possibilities for how SARS-CoV-2 3CL$^{pro}$ induced tau aggregation: (1) both proteins interact and form aggregates; (2) the protease cleaves tau and thus produces insoluble fragments that will initiate the aggregation process. To discover the possible forms of the aggregation endured by tau protein, we performed ThT assays using inactivated 3CL$^{pro}$ by disulfiram (DSF), a known 3CL$^{pro}$ inhibitor [51]. The results demonstrated that inactivation of the protease prevents tau aggregation (Fig 1C), which allows us to suggest that the proteolysis role of the 3CLpro initiates the aggregation process. The addition of DSF in a running ThT assay stopped the tau aggregation immediately (Fig 1D).

## Stability of tau under SARS-CoV-2 3CL$^{pro}$ influence

The stability of 2N4R tau under the influence of SARS-CoV-2 3CL$^{pro}$ was investigated using analytical HPLC experiments. Interestingly, the results demonstrated an evident cleavage of tau protein by the virus protease (Fig 3). As described before, the protein mix was analysed over 0, 24, 48 and 72 h of incubation. The corresponding tau monomer peak (I) (retention time: 10.6 minutes) in the chromatogram decreased over 72h (Fig 3A), which can also be observed in a silver-stained SDS PAGE (Fig 3C). Beside the reduction of the monomer form of tau the presence of a new protein band around 25 kDa was observed, which allow us to suggest that it represents the tau fragments after the 3CL$^{pro}$ proteolytic effect. Additionally, two additional peaks were observed on the analytical HPLC chromatogram (II, III) (Fig 3A, 3D and 3E). SDS PAGE analysis of the related fractions (regions II and III) also validated the increasing protein bands over experimental time.

Based on our results, we assumed that SARS-CoV-2 3CL$^{pro}$ cleaves tau possibly in different sites, resulting in truncated tau species. After 72 h experimental time, the 2N4R tau amount was reduced by about 60% related to the start point. Interestingly, in the first 24 h, the degradation process declines substantially to about 50% of the monomer amount; however, during the remaining 48 h, the monomer amount reduced by just around 10% (Fig 3B). This observation can be attributed to the diminished amount of the monomer itself.

Analytical HPLC with 2N4R tau and inactivated 3CL$^{pro}$ demonstrated that after 72 h, tau is slightly degradated (~5%) (S7 Fig).

## Identification of tau fragments using mass spectrometry

We have used mass spectrometry (MS) experiments to confirm tau fragments in the HPLC peaks I, II and III. Untreated 2N4R tau was used as control; in all tested samples, tau could be detected (Fig 4). The determination of tryptic tau peptides AEPRQEFEVMEDHAGTYGGLGDR, GDTPSLEDEAAGHVTQAR and SPQLATLADEVSASLAK are shown in Fig 4A (S8 and S9 Figs). All identified tryptic tau peptides are listed in S1–S4 Tables (Tryptic peptides that occurred less than three times are not shown).

**Table 2. The number of tryptic tau peptides in control and the tested HPLC samples.**

| Sample | Number tryptic peptides | Peptides with the highest appearance |
|---|---|---|
| 2N4R tau | 111 | AEPRQEFEVMEDHAGTYGGLGDR (8x) |
| I | 66 | SPQLATLADEVSASLAK (8x) |
| II | 46 | GDTPSLEDEAAGHVTQAR (6x) |
| II | 63 | SPQLATLADEVSASLAK (15x) |

In the tau control, 111 tryptic peptides could be determined, and AEPRQEFEVMEDHAGTYGGLGDR had the highest appearance (eight times). Interestingly, we could not identify, in fractions II and III, the N-terminal tryptic peptides detected in the control and peak I (AEPRQEFEVMEDHAGTYGGLGDR and AGLKESPLQTPTEDGSEEPGSETSDAKSTPTAEDVTAPLVDEGAPGKQ) (Fig 4B). Additionally, tryptic peptides in the mid-region and C-terminus of the control sequence could not be detected in fractions related to peak I, II and III (TPPSSGEPPKSGDRSGYSSPGSPGTPGSR, QTAPVPMPDLK and TDHGAEIVYKSPVVSGD). Likewise, in the control sample, the peptide (GSLGNIHHKPGGGQVEVK) was not detected (peak I) (Fig 4B).

Under the experimental conditions (Tau degradation by 3CL$^{pro}$), no fragments are generated that, after tryptic digestion, are amenable of identification, which may influence the composition and number of tryptic tau peptides in the four tested samples, as shown in Table 2.

2N4R tau contains a sequence $_{241}$SRLQTAPV$_{248}$ (QTAPVPMPDLK tryptic peptide absent in peak I) which shows similarities to the preferred 3CL$^{pro}$ cleavage pattern (S10 Fig). A tau cleavage at this site could generate two fragments with sizes of 25 and 20 kDa, shown on the SDS page for peak I (Fig 3C and 3E). Furthermore, at the N-terminus of tau, there are several potential cleavage sites for 3CL$^{pro}$, which can generate fragments with molecular weights between 45 and 27 kDa. Possible 3CL$^{pro}$ cleavage sites relating to the tau protein sequence are shown in S10 Fig; this analysis is based on the amino acid preference in the SARS-CoV-2 3CL$^{pro}$ substrate binding site (Information conceived from the Merops database) [52] and similar amino acid sequences in the tau sequence. Four sequences showed similarities with the 3C-like protease from coronavirus-2 and one with a 3C-like peptidase from strawberry mottle virus (S10 Fig).

The results described in this study indicated that 2N4R tau is proteolytically cleaved by 3CL$^{pro}$, and the cleavage is related to tau aggregation events. It has previously been described that tau proteolysis is associated with aggregation and that the tau protein has cleavage sites for different proteolytic enzymes [34] (Table 3).

**Table 3. Examples of proteases with proteolytic activity against tau.**

| Protease | Cleavage site | Reference |
|---|---|---|
| Caspase-6 | D13-H14 | [53] |
| Caspase-3 | D25-Q26; K44-E45 | [54] |
| Calpain-1 and -2 | R230-T231 | [55, 56] |
| Caspase-2 | D314-L315 | [57] |
| Calpain-1 | K44-E45, R242-L243 | [58, 59] |
| ADAM10 | A152-T153 | [60] |
| Thrombin | R155-G156; R209-S210 | [61] |
| Chymotrypsin | Y197-S198 | [62] |
| Asparagine endopeptidase | N255-V256; N368-K369 | [63] |
| Caspase-1, -3, -6, -7 and -8 | D421-S422 | [64] |

Tau Cleavage sites of Caspase-3 (D25-Q26) and Calpain-1 (R242-L243) are located near possible 3CL$^{pro}$ cleavage sites of tau (Q26-G27 and Q244-T245) (S10 Fig).

## Surface-based fluorescence intensity distribution analysis

Surface-based fluorescence intensity distribution analysis (sFIDA) was performed to quantify the tau oligomers and aggregates after treatment with 3CL$^{pro}$. The technique employs a similar biochemical setup as ELISA-like techniques. However, sFIDA uses the same epitope to capture and detect antibodies and features single-particle sensitivity through a microscopy-based read-out (Herrmann et al., 2017). Recently, sFIDA was applied to quantify tau aggregates in cerebrospinal fluid (CSF) and demonstrated its applicability in clinical settings [37]. Initial sFIDA experiments include analysis of tau monomers, tau aggregates and tau SiNaPs (Fig 5A–5C). To quantify tau aggregates formed by 3CL$^{pro}$ proteolysis, two approaches were tested: tau plus active 3CL$^{pro}$ and tau plus inactivated 3CL$^{pro}$ (with the addition of disulfiram). As shown in Fig 5D, tau samples containing active 3CL$^{pro}$ yielded a similar aggregate-specific readout compared with tau samples in the presence of inactivated protease. Compared to the tau aggregate control, however, only a small fraction of the employed tau substrate was converted into aggregates.

It is well known that the structural diversity of tau aggregates can make their detection technically challenging [65].

For the sFIDA experiments anti-tau12 and anti-tau13 (Biolegend) were used, and both antibodies interact with the N-terminal region of tau (Tau 6–18 and 15–25) [66]. According to the mass spec results described before, tau epitope regions for anti-tau12 and anti-tau13 are cleaved and therefore cannot react with the respective antibodies in the sFIDA assays.

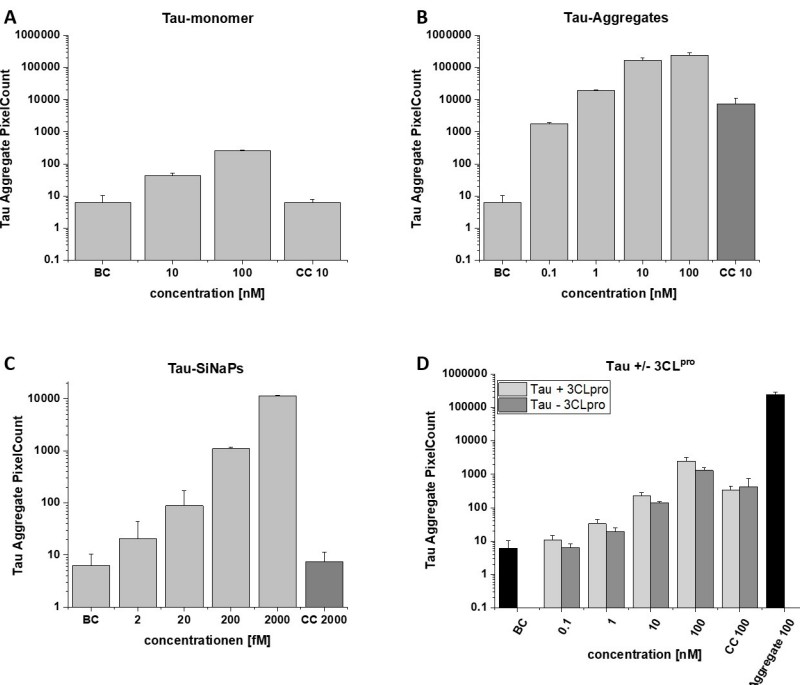

**Fig 5. sFIDA experiments.** Samples, buffer control (BC) and capture control (CC) were tested. Pixel counts per concentration are shown for **A**: Tau monomer control, **B**: Tau aggregate control, **C**: Tau SiNaPs and **D**: Influence of active 3CL$^{pro}$ on tau aggregation.

Additionally, remaining epitopes might be masked by aggregate core formation and can therefore not any more detected by the employed antibody system.

## Conclusion and future work

The proportion of older adults in the population is increasing in almost all countries. Worldwide, around 55 million people have dementia, which is expected to increase to 78 million in 2030 and 139 million in 2050 [67]. Different dementias show a conformationally altered tau, the protein detaches from microtubules and aggregates into oligomers and neurofibrillary tangles, which can be secreted from neurons, and spread through the brain during disease progression.

The COVID-19 pandemic has increasingly moved virus infections into the scientific spotlight and has shown that this infection can damage the brain in many ways. The molecular underpinnings of neurodegenerative processes need to be investigated to develop appropriate therapies. Proteolysis of tau protein may be a crucial factor in forming toxic aggregates. Our results demonstrated that the SARS-CoV-2 3CL$^{pro}$ can cleave 2N4R tau into fragments and thus induce protein aggregation *in vitro*. TEM revealed amorphous aggregates and fibrils; in addition to tau fibrils, structures similar to micelles or vesicles were also observed, serving to nucleate tau fibrillation. However, further experiments such as adjustment and optimisation of sFIDA assay (e.g. antibodies); cleavage and accessibility of tau binding regions for specific antibodies; cell toxicity of tau peptides and related aggregates; *in vivo* experiments to observe the role of the 3CL$^{pro}$ on tau aggregation are needed, to get a closer assessment of the tau cleavage by SARS-CoV-2 3CL$^{pro}$ and if this process plays a pivotal role *in vivo*.

## Supporting information

**S1 Fig. SDS PAGE of 2N4R tau purification.** M: Protein marker; 1: Supernatant after cell disruption; 2: Supernatant after ammonium sulfate precipitation; 3: Pellet after ammonium sulfate precipitation solved in buffer 2; 4: SN after centrifugation pellet solved in buffer 2; 5: Pellet solved in ddH$_2$O and 2 mM TCEP; 6: Pellet was solved in buffer 3; 7: SN after centrifugation pellet solved in buffer 3; 8: Sample after dialysis against buffer 4.
(TIF)

**S2 Fig. Characterization of 2N4R tau. A**: SDS PAGE 15% of 2N4R tau. M: protein marker, 1: pure 2N4R tau after precipitation purification, 2: western blot with CF633 labeled tau13. **B**: CD spectrum of 2N4R tau monomer and filament. **C**: ThT assay of 2N4R tau (10 μM), aggregation was induced with 2.5 μM heparin. Data shown are the mean ± SD from three independent measurements (*n = 3*).
(TIF)

**S3 Fig. Comparison of 2N4R tau and heparin CD spectrum. A**: CD spectrum of 2N4R tau monomer and filament. **B**: CD spectrum of 2.5 μM and 25 μM heparin.
(TIF)

**S4 Fig. SDS PAGE of alpha-synuclein and TDP-43 purification.** M: Protein marker; 1: Supernatant after affinity chromatography; 2: Elution steps after size exclusion chromatography. **A**: SDS PAGE of alpha-synuclein purification. **B**: SDS PAGE of TDP-43 purification.
(TIF)

**S5 Fig. Effect of SARS-CoV-2 3CL$^{pro}$ and tau doses on 2N4R tau aggregation.** The endpoint of the relative fluorescence during a ThT assay is shown. **A**: Effect of different 3CLpro

concentration (0, 0.5, 1, 2.5, 5 and 10 μM) on tau aggregation (Tau concentration was 10 μM). **B**: Effect of different tau concentration (0, 5, 10, 20, 40, 60, 80 and 100 μM) on tau aggregation. (3CL$^{pro}$ concentration was 10 μM). As Control experiments tau at each concentration without protease is shown.
(TIF)

**S6 Fig. Negative stain electron microscopy of 3CL$^{pro}$ induced tau fibrilles. A-F**: Representative micrographs at high (73kx) magnification, showing 3CL$^{pro}$ induced tau fibrilles.
(TIF)

**S7 Fig. Effect of 3CL$^{pro}$ inactivation on tau metabolisation.** The protease was inactivated by 10 μM DSF. **A**: Analytical HPLC analysis of 2N4R tau incubated with inactivated SARS-CoV-2 3CL$^{pro}$ for 0, 24, 48 and 72h. The corresponding chromatogram regions of the tau monomer and related metabolites generated by active 3CL$^{pro}$ are highlighted (I-III). **B**: Stability of 2N4R tau monomer after treatment with inactivated 3CL$^{pro}$ over 72h. The tau monomer amount remains at 95%, compared with a control were tau was treated with active 3CL$^{pro}$. In the control experiment the tau monomer amount reduce to around 40%. **C**: Chromatogram of peak I (Tau) shown enlarged, **D**: Chromatogram of peak region II shown and **E**: Chromatogram of peak region III shown enlarged. Data shown are the mean ± SD from three independent measurements (*n = 3*).
(TIF)

**S8 Fig. Mass spectrometry analysis of tau metabolites (`GDTPSLEDEAAGHVTQAR`).** Example of a MS spectrum corresponding to a tryptic peptide derived from 2N4R tau (`GDTPSLEDEAAGHVTQAR`), b- and y-ions are labelled.
(TIF)

**S9 Fig. Mass spectrometry analysis of tau metabolites (`SPQLATLADEVSASLAK`).** Example of a MS spectrum corresponding to a tryptic peptide derived from 2N4R tau (`SPQLATLADEVSASLAK`), b- and y-ions are labelled.
(TIF)

**S10 Fig. Preferred cleavage sequence pattern of SARS-CoV-2 3CL$^{pro}$ and possible cleavage sites in the tau sequence.** 3CL$^{pro}$ substrate binding site and preferred amino acids are shown and sequence homology of possible cleavage sites. The identical amino acid pattern was checked in the Merops database, if there exist identities to known viral protease cleavage sequences. The following virus proteases were identified: C49.001 (Strawberry mottle virus 3C-like peptidase); C30.001 (Coronavirus picornain 3C-like peptidase-1); C30.003 (Human coronavirus 229E main peptidase); C30.005 (SARS coronavirus picornain 3C-like peptidase) and C30.007 (Coronavirus COVID-19 3C-like peptidase).
(TIF)

**S1 Table. Tryptic peptides of 2N4R tau.**
(PDF)

**S2 Table. Tryptic peptides of 2N4R tau detected in peak I.**
(PDF)

**S3 Table. Tryptic peptides of 2N4R tau detected in peak region II.**
(PDF)

**S4 Table. Tryptic peptides of 2N4R tau detected in peak region III.**
(PDF)

## Acknowledgments

The authors gratefully acknowledge the electron microscopy training, imaging and access time granted by the life science EM facility of the Ernst-Ruska Centre at Forschungszentrum Jülich. We want to thank the support of the Institute of Biological Information Processing (IBI-7) Forschungszentrum Jülich, Germany.

## Author Contributions

**Conceptualization:** Raphael Josef Eberle.

**Formal analysis:** Raphael Josef Eberle, Ian Gering, Anja Stefanski, Victoria Kraemer-Schulien.

**Investigation:** Raphael Josef Eberle, Mônika Aparecida Coronado.

**Methodology:** Raphael Josef Eberle, Mônika Aparecida Coronado, Ian Gering, Simon Sommerhage, Karolina Korostov, Anja Stefanski, Victoria Kraemer-Schulien, Lara Blömeke.

**Resources:** Kai Stühler, Oliver Bannach, Dieter Willbold.

**Supervision:** Raphael Josef Eberle.

**Validation:** Raphael Josef Eberle, Ian Gering, Simon Sommerhage, Anja Stefanski, Victoria Kraemer-Schulien.

**Writing – original draft:** Raphael Josef Eberle.

**Writing – review & editing:** Raphael Josef Eberle, Mônika Aparecida Coronado, Ian Gering, Simon Sommerhage, Karolina Korostov, Anja Stefanski, Kai Stühler, Victoria Kraemer-Schulien, Lara Blömeke, Oliver Bannach, Dieter Willbold.

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
