## [Decision Letter · Decision Letter 0]

19 Apr 2023

PONE-D-23-07828Tau protein aggregation associated with SARS-CoV-2 main proteasePLOS ONE

Dear Dr. Raphael J. Eberle,

Thank you for submitting your manuscript to PLOS ONE. After its careful consideration, we feel that it has merit but does not fully meet PLOS ONE’s publication criteria as it currently stands. Therefore, we invite you to submit a revised version of the manuscript that convincingly addresses the all points raised by both reviewers. Please submit your revised manuscript by July 1st. If you will need more time than this to complete your revisions, please reply to this message or contact the journal office at plosone@plos.org. Please include the following items when submitting your revised manuscript:A rebuttal letter that responds to each point raised by the academic editor and reviewer(s). You should upload this letter as a separate file labeled 'Response to Reviewers'.A marked-up copy of your manuscript that highlights changes made to the original version. You should upload this as a separate file labeled 'Revised Manuscript with Track Changes'.An unmarked version of your revised paper without tracked changes. You should upload this as a separate file labeled 'Manuscript'.

We look forward to receiving your revised manuscript.

Kind regards,

Maria Gasset, Ph.D.

Academic Editor

PLOS ONE

Journal Requirements:

Reviewers' comments:

Reviewer's Responses to Questions

**Comments to the Author**

1. Is the manuscript technically sound, and do the data support the conclusions?

Reviewer #1: Yes

Reviewer #2: Yes

2. Has the statistical analysis been performed appropriately and rigorously? 

Reviewer #1: N/A

Reviewer #2: Yes

3. Have the authors made all data underlying the findings in their manuscript fully available?

Reviewer #1: Yes

Reviewer #2: Yes

4. Is the manuscript presented in an intelligible fashion and written in standard English?

Reviewer #1: Yes

Reviewer #2: No

5. Review Comments to the Author

Reviewer #1: The work presented by Eberle et al. presents novel discoveries of interest to the field of SARS-CoV-2-host interaction, as well as broader implications to the field of neurodegeneration due to the generation of an aggregation-prone truncation of tau by one of the virus proteases.

The authors provide in vitro evidence for a possible triggering mechanism for tau aggregation induced by the SARS-CoV-2 3CL protease (3CLpro), which would cleave full-length tau to generate a pool of fragments, one or more of which would efficiently aggregate. Despite the lack of a more detailed study on the exact cleaving site, the identification of the aggregation-prone fragment, and cell-based studies to better understand the findings, their main claim is justified and supported by the biophysical analysis in the paper. Specifically, Eberle et al. used ThT assays, analytical HPLC and mass spectrometry to report the effect of 3CLpro on tau cleavage and subsequent aggregation. Despite the abovementioned missed points that could improve its appeal to a broader audience, this study provides informative results and interpretations suitable for publication in PLOS ONE, subject to the suggested revisions mentioned below:

Major concerns:

1. The abstract may be misleading as it suggests that the study involves all three proteins (tau, TDP-43, and alpha-synuclein) when, in fact, its focus is only on tau. It would be clearer to specify that this study investigates the effects of 3CLpro on tau aggregation, rather than mentioning TDP-43 and alpha-synuclein which were not affected.

2. The results of this study indicate that 3CLpro proteolytically cleaves 2N4R tau and the cleavage is associated with tau aggregation events. However, it would be beneficial to get a more detailed picture, such as which fragment (or at least which peak observed in HPLC analysis) aggregates, by performing ThT assays on the different fractions obtained upon digestion and HPLC analysis.

Minor concerns:

1. On page 8, line 237, it states that "2N4R tau consists of 266 amino acids with an approximated molecular weight of 46 kDa." However, 2N4R tau is the largest size human brain tau and is composed of 441 amino acids in length; this should be corrected/clarified.

2. Although the analysis of alpha-syn and TDP43 is limited, an SDS-PAGE should be included to assess their sample quality and purity, as done for 2N4R tau in Figure S1.

3. The Western Blotting with CF633 labeled tau13 in Figure S2A (lane 2) should be improved as it shows almost no reactivity.

4. In Figure S2B, a control should be included to show whether heparin on its own contributes to the CD spectrum. Furthermore, the specific method section for CD is missing and should be added, explaining how heparin signal is subtracted from the experimental data during the data processing and analysis.

5. In Figure S2B, the light gray CD spectrum is claimed to correspond to "2N4R tau filament". Imaging analysis (AFM or TEM) should be performed to support this claim.

6. In the legend of Figure 1B (page 9, lines 263-264), it is claimed that "The inactivation of the protease was treated with 10 µM DSF". However, the inactivation condition is reported in Figure 1C, so this should be corrected.

7. On page 9, lines 271-272, it is claimed that "the monomer concentration and its cleavage are accompanied by the amount of aggregate formation (S3B Fig)". However, this observation could be due to the increased concentration of monomer which, even without being cleaved, could aggregate efficiently. In order to support this assertion, the same assay should be performed in the presence of DSF-inactivated protease.

8. In Figure 1D, it is unclear why the fluorescence intensity decreases upon protease deactivation. If aggregates were already formed, the fluorescence intensity should remain unaltered. The decrease could suggest that the formed aggregates can be disassembled or degraded upon inhibitor addition. To clarify this point, a control consisting of aggregated tau (i.e., induced by heparin) + DSF (without protease) should be included to control the potential effect of the inhibitor itself.

9. The two gels shown in Figure 2C should not be cropped and should ideally be shown in a single gel to observe the concomitant decrease of the full-length tau associated with the increase of the 25kDa fragment.

10. As stated in Fig. 4D, "tau samples containing active 3CLpro yielded higher aggregate-specific readouts than tau samples in presence of inactivated protease". However, the readout appears to be quite similar, and a statistical test should be included to support the conclusions.

Reviewer #2: This manuscript describes studies aimed at evaluating the possible induction of protein aggregation by the SARS-CoV-2 3CL protease (3CLpro), which might be relevant in explaining the long term neuropathology induced by infection by this virus.

The Authors performed a series of simple, classic, clear-cut in vitro biochemistry studies based on mixing recombinant tau and 3CLpro and following their interaction using SDS-PAGE and HPLC (to detect fragmentation), or thioflavin fluorescence (to follow aggregation), under different concentration-dependence and kinetic regimens. Mass spectrometry was used to further identify the fragments of tau produced by 3CLpro. Lack of effect of 3CLpro on other proteins relevant to neuropathology, such as synuclein, was also established by similar approaches.

All these experiments were carefully executed, and adequate controls included. Of course, given the high prevalence of neurological ailments within "persistent Covid-19", this line of research is of potential interest.

Having stated this, I find two shortcomings in the manuscript:

1) The ratio of substrate to protease used in these experiments was very high, with a maximum effect when equimolar (10 uM) concentrations were used. Increasing this ratio, to 5, or even 2...resulted in a very substantial decrease of the effect. This casts doubts on the physiological relevance of the results, something that is not acknowledged or discussed by the Authors. Do they envision a set of physiological conditions in which the intracellular or extracellular concentration of 3CLpro are sufficient to cleave tau and cause its aggregation during SARS-Cov-2 infection? Furthermore, the majority of neurological complications of Covid-19 are relatively acute, whereas tau aggregation is involved in neurodegenerative processes (for example, Alzheimer´s disease) which take many years to develop. How do the Authors link a putative 3CLpro-induced aggregation of tau with specific SARS-Cov-2-linked neurological pathology?

2) The manuscript is written in an amateurish fashion (I apologize if the term seems too negative, it is meant to be merely descriptive). A few examples: bullets in the Conclusion section; "Different dementias show a conformationally altered concentration of tau" (a concentration can be high or low, but not "conformationally altered" (line 384); Table 3 in the Discussion, rather than a narrative description of its contents; "...tryptic tau peptides are no longer detectable by 3CLpro into different fragments..." (I understand the Authors mean that 3CLPro, under these experimental conditions, does not generate fragments that, after tryptic digestion, are amenable of identification...)(line 335); Analysis of the tau digestion underwent 3CLpro using mass spectrometrY (????) (line 313); "HPLC experiments with 2N4R tau and inactivated 3CLpro showed that the corresponding tau peak and the peak regions I, II and III are unaffected" (the Authors mean kinetic experiments, HPLC is just an analytical tool, not an experimental approach or design...) (line 310). And many other examples.

Finally, I would suggest analyzing peak I and peak ensembles (they are not "individual peaks") II, and III also directly, withouth prior tryptic digestion. That might allow establishing the size of the different 3CLpro fragments generated, and by combining that information with the sequence of their tryptic fragments (combined, in the case of II and III) their complete characterization.

6. PLOS authors have the option to publish the peer review history of their article (what does this mean?). If published, this will include your full peer review and any attached files.

Reviewer #1: No

Reviewer #2: No

---

## [Author Response · Author response to Decision Letter 0]

8 Jun 2023

Journal Requirements

Comment 1. Please ensure that your manuscript meets PLOS ONE's style requirements, including those for file naming.

Response 1. We checked all submitted files and corrected them accordingly. 

Comment 2. We note that the grant information you provided in the ‘Funding Information’ and ‘Financial Disclosure’ sections do not match. When you resubmit, please ensure that you provide the correct grant numbers for the awards you received for your study in the ‘Funding Information’ section.

Response 2. We corrected accordingly.

Changes by the authors

• We included negative staining Transmission Electron Microscopy (TEM) pictures of the Tau filaments after treatment of 3CLpro. Accordingly we included material and methods, results, new Fig2 and S6 Fig.

• We added one Co-author, Simon Sommerhage, who’s performed the TEM experiments.

Reviewer 1

Comment 1. The abstract may be misleading as it suggests that the study involves all three proteins (tau, TDP-43, and alpha-synuclein) when, in fact, its focus is only on tau. It would be clearer to specify that this study investigates the effects of 3CLpro on tau aggregation, rather than mentioning TDP-43 and alpha-synuclein which were not affected.

Response 1. We changed accordingly

Comment 2. The results of this study indicate that 3CLpro proteolytically cleaves 2N4R tau and the cleavage is associated with tau aggregation events. However, it would be beneficial to get a more detailed picture, such as which fragment (or at least which peak observed in HPLC analysis) aggregates, by performing ThT assays on the different fractions obtained upon digestion and HPLC analysis.

Response 2. We agree with the reviewer's comment. However, preparing the samples for the HPLC analyses includes incubation of tau by 3CLpro, at 37°C for 24h to 72h. After the incubation, we observed significant protein aggregates in the solution, and before applying it to HPLC, we centrifuged the sample to separate aggregates from soluble tau fractions. Consequently, the peak fractions after the HPLC contain soluble tau fragments that are not aggregated. We added a statement in the related material and method section to describe the procedure (lines 181 to 182).

Minor concerns:

Comment 1. On page 8, line 237, it states that "2N4R tau consists of 266 amino acids with an approximated molecular weight of 46 kDa." However, 2N4R tau is the largest size human brain tau and is composed of 441 amino acids in length; this should be corrected/clarified.

Response 1. We corrected accordingly, 266 to 241 amino acids.

Comment 2. Although the analysis of alpha-syn and TDP43 is limited, an SDS-PAGE should be included to assess their sample quality and purity, as done for 2N4R tau in Figure S1.

Response 2. We added SDS pages for TDP43 and alpha-synuclein purification to demonstrate the purity and quality of the proteins. The SDS pages are shown in S4 Fig. 

Comment 3. The Western Blotting with CF633 labeled tau13 in Figure S2A (lane 2) should be improved as it shows almost no reactivity.

Response 3. We added a new western blot in Figure S2A (lane 2).

Comment 4. In Figure S2B, a control should be included to show whether heparin on its own contributes to the CD spectrum. Furthermore, the specific method section for CD is missing and should be added, explaining how heparin signal is subtracted from the experimental data during the data processing and analysis.

Response 4. We added the CD spectrum of 2.5 µM heparin, which showed no signal in the measured wavelength range (190-240 nm). As described in the material and methods section, this heparin concentration was used to induce tau aggregation. We also added a CD spectrum of a tenfold higher heparin concentration (25 µM), which demonstrated a minimum in the CD spectra at 220 nm. However, the heparin concentration used for tau aggregation (2.5 µM) showed no signal on the CD-spectrum, demonstrated in the new S3 Figure and a statement in the results lines 284 to 287. Additionally, we added the missing CD method section. 

Comment 5. In Figure S2B, the light gray CD spectrum is claimed to correspond to "2N4R tau filament". Imaging analysis (AFM or TEM) should be performed to support this claim.

Response 5. It is well established that CD can be used to follow tau aggregation as the process accompanies a transition from an unfolded structure to a partially folded structure with approximately 36% of β-sheet. The CD spectra show this typical shift from an unfolded structure (around 200 nm) to a structure containing β-sheet (around 220). 

This observation is described in various publications, and a few examples is mentioned in the reference list:

• https://doi.org/10.1021/bi0357006

• https://doi.org/10.1016/j.bbadis.2004.09.010

• https://doi.org/10.1016/j.pep.2016.09.009

Comment 6. In the legend of Figure 1B (page 9, lines 263-264), it is claimed that "The inactivation of the protease was treated with 10 µM DSF". However, the inactivation condition is reported in Figure 1C, so this should be corrected.

Response 6. We corrected accordingly

Comment 7. On page 9, lines 271-272, it is claimed that "the monomer concentration and its cleavage are accompanied by the amount of aggregate formation (S3B Fig)". However, this observation could be due to the increased concentration of monomer which, even without being cleaved, could aggregate efficiently. In order to support this assertion, the same assay should be performed in the presence of DSF-inactivated protease.

Response 7. Control experiments were performed with different tau concentrations in the absence of the protease to demonstrate that there is no concentration-dependent in the aggregation process. We added the control experiments to Figure S5B. 

Comment 8. In Figure 1D, it is unclear why the fluorescence intensity decreases upon protease deactivation. If aggregates were already formed, the fluorescence intensity should remain unaltered. The decrease could suggest that the formed aggregates can be disassembled or degraded upon inhibitor addition. To clarify this point, a control consisting of aggregated tau (i.e., induced by heparin) + DSF (without protease) should be included to control the potential effect of the inhibitor itself.

Response 8. Thanks for this comment. We corrected it accordngly.

Comment 9. The two gels shown in Figure 2C should not be cropped and should ideally be shown in a single gel to observe the concomitant decrease of the full-length tau associated with the increase of the 25kDa fragment.

Response 9. Fig 2C was changed accordingly.

Comment 10. As stated in Fig. 4D, "tau samples containing active 3CLpro yielded higher aggregate-specific readouts than tau samples in presence of inactivated protease". However, the readout appears to be quite similar, and a statistical test should be included to support the conclusions.

Response 10.

We changed the sentence accordingly. New statements were written in the results section, “Compared to the tau aggregate control, however, only a small fraction of the employed tau substrate was converted into aggregates”. We also mentioned in conclusion, “Adjustment and optimisation of sFIDA assay need to be performed to get meaningful results using this technique. 

We changed the questionable sentence to “tau samples containing active 3CLpro yielded a similar aggregate-specific readout compared with tau samples in the presence of inactivated protease”.

Reviewer 2

Comment 1. The ratio of substrate to protease used in these experiments was very high, with a maximum effect when equimolar (10 uM) concentrations were used. Increasing this ratio, to 5, or even 2...resulted in a very substantial decrease of the effect. This casts doubts on the physiological relevance of the results, something that is not acknowledged or discussed by the Authors. Do they envision a set of physiological conditions in which the intracellular or extracellular concentration of 3CLpro are sufficient to cleave tau and cause its aggregation during SARS-Cov-2 infection? Furthermore, the majority of neurological complications of Covid-19 are relatively acute, whereas tau aggregation is involved in neurodegenerative processes (for example, Alzheimer´s disease) which take many years to develop. How do the Authors link a putative 3CLpro-induced aggregation of tau with specific SARS-Cov-2-linked neurological pathology?

Response 1. We agree with the reviewer. However, this manuscript intended to demonstrate the potential of SARS-CoV-2 3CLpro to cleave tau in vitro and that this cleavage can induce aggregation of tau under the tested conditions. We need more experiments to get information about the physiological relevance and involvement in neurodegeneration. We are not claiming that SARS-CoV-2 3CLpro can induce neurodegenerative processes in vivo after a SARS-CoV-2 infection. For such a statement, we need specific experiments not conducted so far. We changed the conclusion accordingly to state that more experiments are necessary under physiological conditions and for an extended period. 

However, two studies could connect COVID-19 infections with neurodegenerative processes:

• https://doi.org/10.3390/cells10020386

• https://doi.org/10.15252/embj.2020106230

We mentioned both studies in the introduction (References 30 and 33). 

Comment 2. The manuscript is written in an amateurish fashion (I apologise if the term seems too negative, it is meant to be merely descriptive). A few examples: bullets in the Conclusion section; "Different dementias show a conformationally altered concentration of tau" (a concentration can be high or low, but not "conformationally altered" (line 384); Table 3 in the Discussion, rather than a narrative description of its contents; "...tryptic tau peptides are no longer detectable by 3CLpro into different fragments..." (I understand the Authors mean that 3CLPro, under these experimental conditions, does not generate fragments that, after tryptic digestion, are amenable of identification...)(line 335); Analysis of the tau digestion underwent 3CLpro using mass spectrometrY (????) (line 313); "HPLC experiments with 2N4R tau and inactivated 3CLpro showed that the corresponding tau peak and the peak regions I, II and III are unaffected" (the Authors mean kinetic experiments, HPLC is just an analytical tool, not an experimental approach or design...) (line 310). And many other examples.

Response 2. Several changes were made in the manuscript accordingly to reviwers suggestions.

Comment 3. Finally, I would suggest analysing peak I and peak ensembles (they are not "individual peaks") II, and III also directly, without prior tryptic digestion. That might allow establishing the size of the different 3CLpro fragments generated, and by combining that information with the sequence of their tryptic fragments (combined, in the case of II and III) their complete characterisation.

Response 3. The mass spectrometric experiments of the peaks aimed to confirm that they contain different tau species (Fragments) generated by the 3CLpro activity. As mentioned, it is the first finding on the role of the main protease in tau cleavage and aggregation. We will undoubtedly perform new experiments to obtain a detailed view of the tau fragments.

---

## [Decision Letter · Decision Letter 1]

20 Jun 2023

Tau protein aggregation associated with SARS-CoV-2 main protease

PONE-D-23-07828R1

Dear Dr. Raphael J. Eberle,

We’re pleased to inform you that your manuscript has been judged scientifically suitable for publication and will be formally accepted for publication once it meets all outstanding technical requirements.

Kind regards,

Maria Gasset, Ph.D.

Academic Editor

PLOS ONE

Additional Editor Comments (optional):

Reviewers' comments:

Reviewer's Responses to Questions

**Comments to the Author**

1. If the authors have adequately addressed your comments raised in a previous round of review and you feel that this manuscript is now acceptable for publication, you may indicate that here to bypass the “Comments to the Author” section, enter your conflict of interest statement in the “Confidential to Editor” section, and submit your "Accept" recommendation.

Reviewer #1: All comments have been addressed

2. Is the manuscript technically sound, and do the data support the conclusions?

Reviewer #1: Yes

3. Has the statistical analysis been performed appropriately and rigorously? 

Reviewer #1: N/A

4. Have the authors made all data underlying the findings in their manuscript fully available?

Reviewer #1: Yes

5. Is the manuscript presented in an intelligible fashion and written in standard English?

Reviewer #1: Yes

6. Review Comments to the Author

Reviewer #1: I am delighted to announce that I have thoroughly reviewed the submitted manuscript and all my comments and concerns have been addressed by the authors. After consideration, I recommend the publication of this paper in PLOS ONE. The authors have diligently responded to the reviewers' feedback, implementing necessary revisions and improvements to enhance the quality and clarity of their work.

7. PLOS authors have the option to publish the peer review history of their article (what does this mean?). If published, this will include your full peer review and any attached files.

Reviewer #1: No

---

## [Editor Report · Acceptance letter]

11 Aug 2023

PONE-D-23-07828R1 

Tau protein aggregation associated with SARS-CoV-2 main protease 

Dear Dr. Eberle:

I'm pleased to inform you that your manuscript has been deemed suitable for publication in PLOS ONE. Congratulations! Your manuscript is now with our production department. 

Kind regards, 

on behalf of

Dr. Maria Gasset 

Academic Editor

PLOS ONE